# Estimating the effects of crime maps on house prices using an (Un)natural experiment: A study protocol

**Meng Le Zhang**[1]*, **Monsuru Adepeju**[2], **Rhiannon Thomas**[1]

**1** Department of Social Science, University of Sheffield, Sheffield, United Kingdom, **2** Department of Sociology, Manchester Metropolitan University, Manchester, United Kingdom

* meng_le.zhang@sheffield.ac.uk

## Abstract

Street-level crime maps are publicly available online in England and Wales. However, there was initial resistance to the publication of such fine-grained crime statistics, which can lower house prices and increase insurance premiums in high crime neighbourhoods. Identifying the causal effect of public crime statistics is difficult since crime statistics generally mirror actual crime. To address this question empirically, we would ideally experiment and introduce a source of random variation in the crime statistics. For instance, we could randomly increase or decrease the number of offences displayed in crime statistics and measure their effects on local house prices. For obvious reasons, we cannot pursue this research design. However, street-level crime maps contain intentional errors, which are the product of a geomasking algorithm designed to mask the location of crimes and protect the identity of victims. This project leverages features associated with the geomasking algorithm to estimate the effect of public crime statistics on house prices.

## Introduction

Crime may affect house prices through mediating causal pathways–such as the destruction of property or victimisation of locals [1, 2]. One mediating pathway is the 'signalling' effect of crime which may decrease house prices in high crime areas due to a perception of increased victimisation or other factors like undesirable neighbours. The public may form their opinions about crime from several sources, from word of mouth to official statistics. Since 2011, the UK government has published online monthly crime figures at almost street-level resolution [3]. On launch in February 2011, the crime map website (henceforth referred to by its domain name police.uk) received over 18 million visits an hour which caused the website to crash repeatedly [4]. Before the publication of online crime maps on police.uk, street-level crime data was never available to the public (except in West Yorkshire [3]).

The release of online crime maps was supposed to create public confidence in crime statistics and support public service transparency [5]. The publication of these maps was met by opposition who feared it would affect house prices and increase insurance premiums, particularly in high crime low-income neighbourhoods [4, 6]. At the time, the UK government argued that the benefits of open public service data outweighed these concerns [7]. However, the

**Data Availability Statement:** The HM Land Registry Price Paid dataset is a publicly available dataset of properties sold in England and Wales since 1995. The dataset contains information on price sold, address, house type and other features

of the property sold. The dataset excludes certain types of transactions, such as inheritance and discounted transactions (e.g. discounted sales of social housing under the 'Right to Buy' scheme). The dataset can be accessed at: https://www.gov.uk/government/collections/price-paid-data. The coordinates of a property are derived from the coordinates of its postcode as recorded in the ONS National Statistics Postcode Lookup (ONS, 2021). (https://geoportal.statistics.gov.uk/datasets/7606baba633d4bbca3f2510ab78acf61/about). Archival data from police.uk are publicly available from the police.uk data site (https://data.police.uk/data/archive/). Other information, such as police force boundaries, are also contained on the website. We use the earliest archival extract of police.uk which contains data on crimes from Dec 2010 to Dec 2013.

**Funding:** This project is funded by a British Academy/ Leverhulme small grant (SRG21\210192, awarded to MZ and MA). Funder website: https://www.thebritishacademy.ac.uk/funding/ba-leverhulme-small-research-grants/ The funder had and will not have a role in study design, data collection and analysis, decision to publish, or preparation of the manuscript.

**Competing interests:** The authors have declared that no competing interests exist.

advantages and disadvantages of publicly available crime data as a policy are unclear. Many countries with similar capabilities do not make such information public. For example, Scotland has not followed the rest of the UK in this respect.

In this project, we are interested in the causal effect of public crime maps on house prices. This effect is separate from the impact of crime on house prices through other causal pathways such as damage to the environment or negative public perceptions due to victimisation in the local area. The effect of crime maps on house prices is difficult to identify since accurate crime statistics would perfectly mirror actual crime. It is impossible to estimate the effects of crime statistics from actual crime separately in this scenario.

To address this question empirically, we would ideally conduct an experiment and introduce a source of variation in the crime statistics. For instance, across England and Wales, we could randomly increase or decrease the number of offences displayed on police.uk for a period to measure the effects. For obvious reasons, we cannot pursue this research design. However, intentional errors introduced into police.uk crime maps through geomasking processes can be leveraged as a source of variation to investigate the signalling effect of the crime maps on the house prices.

Police.uk implements a geomasking algorithm that obscures the actual location of crimes in order to protect the identity of victims. The geomasking algorithm allocates crimes to a nearby geographical location called a snap-point [6, 8]. Whilst the density of snap-points in urban areas is very high, the level of crime in a small area measured using police.uk can differ substantially from actual police records due to geomasking [8]. This error level gets progressively worse at smaller spatial scales. In 80% of postcodes, the local area crime counts as recorded by police.uk is substantially different from that measured by actual police sources [8]. In short, local area crime statistics on police.uk can be considerably lower or higher than what they should be. This mismatch (henceforth the geomasking error) constitutes one source of variation that we will use in our (un)natural experiment.

The geomasking routine produces another key map feature associated with police.uk: the number of potential snap-points in an area (i.e. around a house). Another test of whether police.uk affected house prices is to compare the statistical association between snap-points before and after the launch of street-level crime maps. A key part of this study is that the secret list of snap-points used by police.uk can be inferred using public data alone.

## Research questions and hypotheses

The primary research questions are:

**RQ1.** Did police.uk crime statistics affect property prices? We test this indirectly using snap-point data only.

**RQ2.** What is the effect of a one-unit increase in crime around a house (as reported by crime maps) on its selling price? We test this directly using geomasking errors.

We have the following hypotheses:

**H1.** The null hypothesis related to RQ1 is that police.uk did not affect house prices. This was the position of the Home Office and some property analysts in 2011 following criticisms of the website [7]. Our hypothesis is that police.uk lowered house prices in high crime areas. This was the position of other property analysts and estate agents in 2011 [4].

**H2.** The null hypothesis related to RQ2 is that the number of crimes shown on police.uk will not decrease house prices. This was the position of the Home Office [7]. Our hypothesis is that an increase in the number of crimes shown on police.uk's crime map in an area should lower house prices. This is in line with economic theory that perceptions of high crime make houses less desirable.

The objective of this study protocol is to specify the research plan ahead of data collection/ access. For RQ1, we will use public domain data only and examine evidence from all 43 police forces in England and Wales. RQ2 cannot be answered without access to the original geocoded crime data used by police.uk in their online crime maps. To this end, we have gained the support of the South Yorkshire Police (henceforth SYP) Force to access and use their geocoded crimes and incidence data.

## Previous studies

To our knowledge, no other study has studied the effect of crime maps on actual house prices or leveraged the same research design. However, there are related papers about the effects of crime maps on public perception [9–13]. We exclude four older studies (pre-2000s) mentioned in [9] looking at the introduction of crime statistics in general. These studies employ experimental designs comparing survey respondents' reactions to different types of crime maps (dots versus density/ hotspots) or crime data (maps versus tabular data). The one exception is [11], which is an exploratory qualitative evaluation. These study are low in size (n < 200) with the exception of [10] (n = 7434). There is limited evidence to suggest that crime maps do not cause additional fear of crime compared to tabular data [9, 10]. Wuschke et al. [13] compared two types of crime maps and found that participants perceived higher crime incidents on density maps compared to dot maps (both using the same underlying data). Strangely, participants also thought house prices were higher in the density maps. Whilst these experimental studies have strong internal validity, they suffer from weak external validity. It is unknown whether findings from survey responses to questions about house prices or perceptions of neighbourhood quality can be extended to the actual housing market [10]. Furthermore, due to their recruitment strategies, the participants are not usually representative of the wider population or the population of home buyers [9, 13].

On the other hand, several non-experimental studies have explored the relationship between crime in general (not crime maps) and house prices. UK-based studies have found that areas with higher crime also have lower house prices [2, 14]. Ihlanfeldt and Mayock [15] reviewed 18 hedonic price studies that included a measure of neighbourhood crime among the explanatory variables. Generally, crime indicators are negatively associated with house prices. However, as pointed out in their review, the majority of studies lack a credible research design to deal with unobserved characteristics associated with neighbourhood crime and house prices (confounders). The majority rely on statistical adjustment alone with no ability to test for bias from omitted variable bias.

Whilst we do not know of other studies with the same research design, our general strategy of leveraging differences between what is publicly seen and what is actually the case is well-established in economics. In labour economics, coarsened information about overall course grades has been used to study the 'signalling' effects of education on labour market outcomes [16]. Coarsened public information about Radon has been used to study the effects of Radon on house prices [17].

The existence of individual police force crime maps before police.uk has been documented in several studies [3, 18]. Quinton [10] trialled the effects of crime maps as a pilot before the launch of police.uk. The trial involved very aggregate level crime maps, and participants seemed to spend very little time perusing them (average, ~50 seconds). The primary outcome was perceptions of crime. Police.uk's geomasking algorithm has been studied in several papers [5, 8, 19, 20]. Tompson et al's paper [8] on how geomasking errors were substantial in lower geographies (e.g. postcodes) greatly influenced this project. Finally, information about the data on police.uk is published on a companion data site and past versions are available from

internet archives [21]. The development of the site is also documented in an academic paper [19]. This project would not have been possible without the excellent documentation in these sources.

## Materials and methods

### Research design

Members of the public or other bodies (e.g. estate agents) are not aware of the actual location of crimes as recorded by the police. Also, street-level crime maps for South Yorkshire did not exist before police.uk. Broader areal level maps have existed since 2008; notably the *crimemapper* website. The research design exploits the fact that key features of police.uk, namely its snap point list and geomasking errors, did not exist before 2011 and had no causal impact on house prices before that date. Any statistical association between these features and house prices are due to the existence of confounders: common causes that affect both police.uk crime map features and house prices. After the launch of police.uk in 2011, these crime map features could have a causal effect on house prices. Assuming that the relationship between confounders and these map features are constant over time, changes in the association between these map features and house prices before and after the launch of police.uk will be indicative of causality. In essence, we are employing a form of interrupted time series analysis. Fig 1 demonstrates the intuition behind our research design.

First, we can test for the impact of police.uk on house prices if we knew the secret list of snap-points used by police.uk. These snap-points are based on a subset of primarily static urban features (mainly the centre of residential roads) as recorded in 2012 (and maybe 2011). Before 2011, the number of potential snap-points around a house could not casually affect its

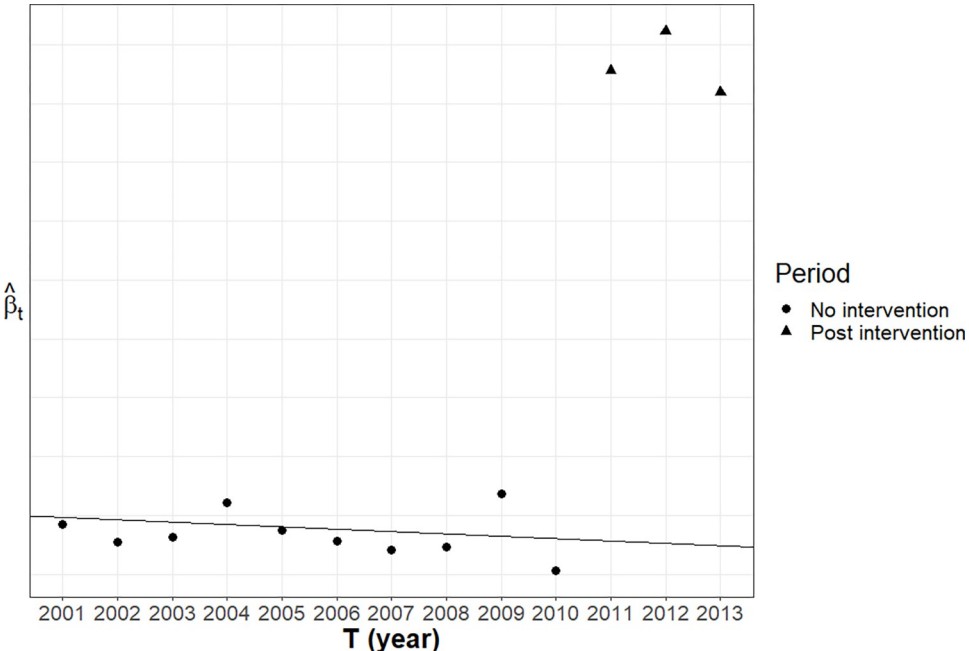

**Fig 1. Example of an interrupted time series using police.uk data.** $\hat{\beta}_T$ is the associations (e.g. correlation coefficient, regression slope) between house prices and either number of snap-point or geomasking error. A change in $\hat{\beta}_T$ after the launch of police.uk in 2011 indicates causality. We can also check for a constant trend in $\hat{\beta}_T$ before the intervention (e.g. $\hat{\beta}_T$ doesn't wildly fluctuate for no reason).

selling price. It will have an association with selling price because the snap-points are constructed from Ordinance Survey data which in turn reflect urban features (e.g street centres). Both could have different effects on house prices. After the launch of police.uk, the number of potential snaps around a house could have a causal effect on house prices as areas with more snaps will have more crimes shown. For example, at the extreme end, houses in areas with no potential snap-points nearby will have no crimes reported on the police.uk crime map. Police.uk could have an effect outside of the information shown on its crime maps. Most notably, it could affect the credibility of other sources of crime information. For instance, more accurate crime reporting on the website could reduce the impact of other sources such as word of mouth or older crime maps. On the other hand, errors in the location of crimes on police.uk weakens the relationship between crimes and house prices.

Second, using snap-points alone, we cannot disentangle the impact of crime shown on crime maps from the effect of police.uk on other causal factors. We will use geomasking error around a property as a source of variation in the crime shown on a crime map to resolve this issue. If the level of geomasking error was completely random, we could use it to estimate the effects of police.uk on house prices. However, geomasking errors are not entirely random. Due to the rule governing the creation of the snap-points, high levels of geomasking error are associated with proximity to urban features (e.g. high density housing, centre of roads). These features will also affect house prices.

As stated above, these urban features are primarily static over time and space, and therefore their confounding effects tend to remain invariant over time. Furthermore, the geomasking technique used by police.uk can be replicated since it is well documented and based on a list of static snap-points. Therefore, we can mimic what public crime information would have looked like on the website if it had launched a year or two earlier. Then we can calculate geomasking errors around an area for periods before police.uk's launch.

As with potential snap-points, we would look for differences in the association between geomasking errors and in periods before and after police.uk's launch. Our preferred estimator would compare houses sold in areas with different levels of geomasking error but the same level of real crime. This is to block other causal pathways between police.uk and house prices that are not related to the information shown on the crime maps (see Statistical Analysis section).

## Sample

Our unit of analysis are residential properties that were sold in England and Wales. We will cover a period from 2010 (or earlier) to 2013. We further restrict our sample to properties:—which are not newly built—whose Price Paid Data (PPD) category is category A (see data section)—not transfered as part of a discounted sale (e.g. discounted social housing) or inheritance—not in the top or bottom 2% of price sold that year (trimming outliers).

For RQ1, we examine data from every police force (n = 43) within England and Wales. The earliest date covered by our data is 1995.

For RQ2, we restrict the sample to all properties sold within South Yorkshire Police's force boundaries. We do not know the earliest date covered by SYP data (which is needed to calculated geomasking errors). We have been told that SYP data from at least 2010 is available. Using public data sources (see Data section), in South Yorkshire we have roughly 10,000 properties in our eligible sample every year between 2010 and 2012 (12,000 sold in 2013). The mean house price is virtually unchanged while the price variance increased in 2013 (see Table 1).

The year 2013 was chosen as the end period for several reasons. First, we want to investigate in detail what happened in the years around the launch of police.uk. Given our small resources,

**Table 1. Summary statistics for South Yorkshire only.** All prices are in GBP.

| Year | N | Mean Price | Mean Log Price | Sd Price | Sd Log Price |
|------|------|------------|----------------|----------|--------------|
| **2010** | 9598 | 143983 | 11.7 | 94415 | 0.530 |
| **2011** | 9798 | 135934 | 11.7 | 96959 | 0.521 |
| **2012** | 9915 | 141522 | 11.7 | 94802 | 0.532 |
| **2013** | 12064 | 142829 | 11.7 | 118282 | 0.546 |

we have to constrain ourselves to a few years to investigate. Second, we want to make sure that the housing market and policy remain relatively stable in the years just before and after police. uk. Three years after the launch of police.uk seemed to be sensible. Third, in 2013 a scheme aimed at helping first time home buyers called Help to Buy was launched. The first phase of the scheme only affected buyers looking to buy newly built homes (which are excluded from our sample). The second phase was launched in October 2013 and did not just affect newly built homes. This scheme increased demand in the housing market, but it's not apparent how that will affect the validity of our design. As a precaution, we limit our protocol analysis to years up to 2013 (i.e to balance credibility and statistical power).

To do a power analysis, we need to know the variance of the 'treatment' variable: the sum of potential snap-points for RQ1 ($M_S$) and the crime counts on police.uk (conditional on real crime counts) for RQ2. For RQ2, we cannot know this statistic before accessing the data. For RQ1, we have ran our statistical analysis for one police force only (South Yorkshire, see S1 File). From those results, we are confident in the statistical power of our estimator when applied to other police force in England and Wales.

## Statistical analysis

Begin with a limited version of the causal relationship between snap-points around a house, crimes on police.uk around a house, selling price and confounders (many of which we observe). Let:

- $Y$ selling price of a house (logged). $Y$ can be demeaned to adjust for inflation but this is irrelevant later (e.g. due to the inclusion of an intercept in OLS models).

- $C_g$ Total crime counts around a house using police.uk (i.e. geomasked crime count). For records before December 2010, we use South Yorkshire Police's geocoded data, our inferred-snap list, and details from data.police.uk [22] to create what data would have been on police. uk if it had launched earlier (see data section).

- $C_r$ Total crime counts around a house using police force records. Although errors can exist in the police data, we assume this is the real crime count for simplicity. We do not believe this will adversely affect our design.

- $M_s$ Sum of snap-points around a house

- $U$ confounding factors affecting $Y$ and other variables.

  Examples of confounders are:

- The real location of crimes which can lower house prices in an area. Higher areas of crime could also result in higher geomasking error.

- The locations of houses sold. Location can affect house prices and we know that snap-points are located near urban features such as the centre of roads.

Imagine two period: one before the launch of police.uk ($T = 0$) and one afterwards ($T = 1$). For now, assume that $T = 0$ refers to the year 2010 and $T = 1$ is the year 2011. Furthermore imagine these counterfactuls: let $W = 1$ be a world where police.uk's website existed during period $T$ and $W = 0$ be a world it didn't.

We know for certain that key features of police.uk, namely snap-points and crimes shown ($M_s$ and $C_g$), has no causal impact when $W = 0$. If the website did not exist–its contents cannot affect house prices. When $W = 0$, the statistical association between $M_s$ and $C_g$ due to confounder(s) $U$. To discover whether police.uk affected house prices, we compare the statistical association between i) $M_s$ and $Y$ and ii) $C_g$ and $Y$. For example, we can check if:

$$P(Y|M_s, T = 1, W = 1) - P(Y|M_s, T = 1, W = 0)$$

However, we cannot ever observe a world in which police.uk did not exist in 2011 (i.e. $T = 1$, $W = 0$). But we do observe data from the year 2010 when police.uk did no exist (i.e. $T = 0$, $W = 0$). We can substitute data from 2010 for the data from the counterfactual $T = 0$, $W = 0$.

The below Directed Acyclic Graph (DAG) (Fig 2) represents our core assumptions about causal relationships in a world where police.uk did not exist [23]. A more extensive version is shown in S2 File.

**Estimators (RQ1).**   From the DAG, at $T = 0$ we can see that the statistical correlation between $Y$ and $M_s$ is simply due to confounding. If the police.uk affected house prices, then this correlation will change at $T = 1$. Therefore we can answer RQ1 by checking the statistical relationship between house prices and the number of potential nearby snap-points $M_s$.

*Estimator 1A*: *Non-parametric*. Calculate the correlation (i.e. spearman's rank) between $Y$ and $M_s$ at $T = 0$ and $T = 1$ separately. Under the null hypothesis the correlation would be identical for both time periods. We can use bootstrapping or permutation tests to get standard errors.

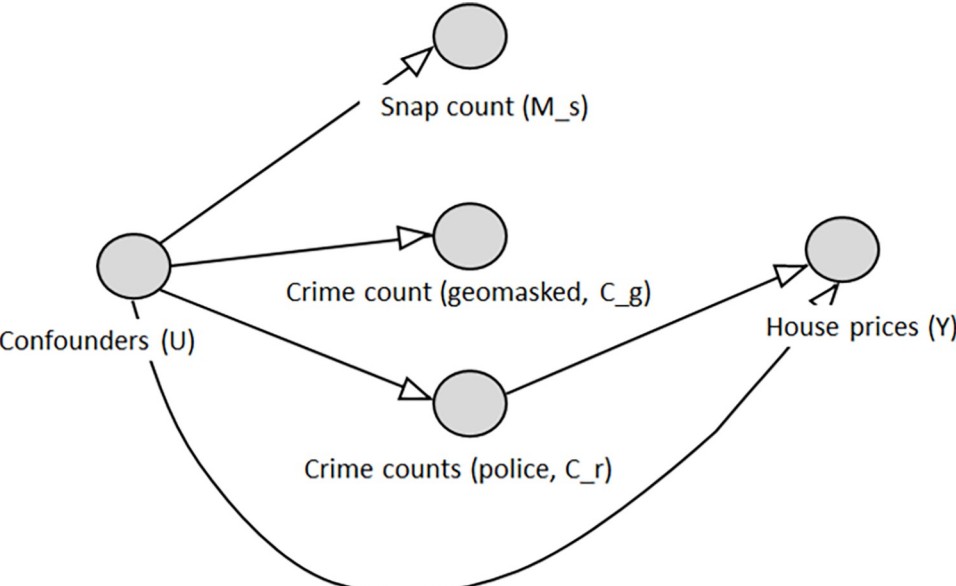

**Fig 2. DAG/ path diagram of causal relations.** We assume the existence of police.uk only affected house prices through i) the information shown on the website (modifying $C_g{\rightarrow}Y$) and ii) changing the relationship between information outside the website and house prices ($U{\rightarrow}Y$, $U{\rightarrow}C_r{\rightarrow}Y$). Other causal relationships are unchanged, and this forms the basis for our assumption tests later.

*Estimator 1B*: *Parametric*. Assuming that a linear functional form is close enough approximation, estimate the following model using OLS:

$$E(Y) = \beta_0 + \beta_T T + \beta_{M_s} M_s + \beta_{TM_s} T.M_s$$

Where $\beta$ denotes parameters to be estimated from the data. The interaction term $\beta_{TM_s}$ denotes a change in the statistical relationship between $M_s$ and $Y$ over time. Under the null hypothesis that police.uk had no effect, we would expect $\beta_{TM_s}$ to be zero.

*Estimator 1C*: *Parametric time series*. Assume that a linear functional form is close enough approximation. Then redefine $T$ to be an index number for year (e.g. $T = 2011$ if the year is 2011). For every year, estimate the following model using OLS:

$$E(Y_T) = \beta_{0,T} + \beta_{M_s,T} M_s$$

The statistical association between $Y$ and $M_s$ $(\beta_{M_s,T})$ may change over time. If that change was consistent (e.g. greater effect sizes due to inflation) then, for every year before 2011, we can fit the following model using OLS:

$$E(\beta_{M_s,T}) = \delta_0 + \delta_T T$$

We can use the model to estimate $\tilde{\beta}_{M_s,2011}$. This is what $\beta_{M_s,2011}$ would have been if police.uk did not launch in 2011. Under the null hypothesis:

$$\beta_{M_s,2011} - \tilde{\beta}_{M_s,2011} = 0$$

**Estimators (RQ2).** To answer RQ2, we can directly estimate the effect of the information shown on its website $C_g$ by exploiting the discrepancies in $C_g$, which is what the public observes, and the actual crime count $C_r$. By comparing areas with the same underlying crime levels ($C_r$) but different crime counts as shown on police.uk, we can estimate the causal effect of $C_g$ on house prices.

This is only possible if either police.uk's had no effect on house prices outside its website ($U{\rightarrow}Y$) or police.uk only affected the impact of other crime-related factors on house prices. For instance, if police.uk affected the credibility of our sources of crime information like word of mouth or caused the shutdown of other crime maps. In this instance, we assume that this is entirely captured by the causal relationship $C_r{\rightarrow}Y$ and that other pathways $U{\rightarrow}Y$ are unchanged.

Under such conditions, we can compare prices for houses sold in areas with the same underlying crime levels ($C_r$) but different crimes counts ($C_g$) as shown on police.uk. From the DAG, the observed statistical relationship between $C_g$ and $Y$ conditional on $C_r$ is a function of both confounding and collider bias ($C_r$ is a collider). If $C_g$ had no effect on house prices then this statistical relationship should remain exactly the same at $T = 0$ and $T = 1$.

*Estimator 2A*: *Non-parametric*. Check $E(Y|C_g, C_r, T)$: the expected value of $Y$ given $C_g$, $C_r$ and $T$. Since $C_g$ and $C_r$ are continuous counts, we will have to band these into categories (based on say quartiles). For example, a house is in band $B_1$ if its values of $C_g$ and $C_r$ fall in the lowest quartile.

For each band, calculate difference mean $Y$ over time:

$$E(Y|B_b, T = 1) - E(Y|B_b, T = 0)$$

Under the null hypothesis, this value should be equal to zero. Given the large number of potential bands, we may have to correct for multiple tests (e.g. Bonferroni correction).

*Estimator 2B*: *Semi-parametric using matching*. Match each case (i.e. house sold) in $T = 1$ to cases in $T = 0$ based on $C_r$ and $C_g$. Discard unmatched cases. Ideally, the remaining data in $T = 0$ and $T = 1$ should have identical distributions of $C_r$ and $C_g$ (i.e. $P(C_g, C_r|T) = P(C_g, C_r)$).

Assuming approximate linearity, estimate the following using OLS:

$$E(Y) = \beta_0 + \beta_T T + \beta_{C_g} C_g + \beta_{TC_g} T.C_g + \beta_{C_r} C_r + \beta_{TC_r} T.C_r$$

Under the null hypothesis that $C_g$ has no effect, $\beta_{TC_g}$ is equal to zero. $\beta_{TC_g}$ is the estimate for the impact of a one-unit increase in $C_g$ on house prices. We can also replace $C_r$ with dummy variables for $C_r$ quartiles.

**Alternative estimators.**   We can further extend estimators 1B, 1C, and 2B using areal fixed effects. We will use lower super output area (LSOA) fixed effects. This is because public information at levels lower than LSOA is rare due to small area estimation and privacy issues. For example, the much-used index of multiple deprivation only reports deprivation at the LSOA level. We are almost certain that the older crime maps did not contain information at a lowest geographical scale than LSOA (with one potential exception for West Yorkshire). In addition, fixed effects control for time-variant confounders. As we mentioned previously, the major sources of confounding are due to static urban features and arise due to how police.uk created the snap-point list.

For example, the alternative estimator with fixed effects (FE) for 1B would be:

$$E(Y) = \beta_0 + \beta_T T + \beta_{M_s} M_s + \beta_{TM_s} T.M_s + F_{LSOA}$$

Where $F_{LSOA}$ are the LSOA fixed effects. The quantity of interest (i.e. $\beta_{TM_s}$) and all other things remain unchanged. Standard errors have to be adjusted for fixed effects.

Instead of using the log of house prices as our outcomes $Y$, we can try to use house prices without any transformations. This is only a useful estimator if inflation between $T = 0$ and $T = 1$ is negligible. Else if inflation is non-negligible but consistent, then estimator 1C (i.e. time series) will remain unbiased.

Another alternate estimator uses the relationship between crimes shown on police.uk $C_g$ and $Y$ to answer RQ1. Replace $M_s$ with $C_g$ in estimators 1A - 1C and change the relevant quantities of interest. Everything else remains the same. There are data limitations based on how many years of geocoded data (before 2011) have been archived by police forces.

For estimator 1C, which is relevant in case of trends over time, we can explore more elaborate time series models with different techniques for identifying structural breaks.

Our statistical tests are designed to reject the null hypothesis. Failure to reject the null hypothesis does not mean that the null hypothesis is true (i.e. police.uk did not affect house prices). An alternative approach is to calculate the probability that the true effect size is higher than some substantial amount. For example, if we can check the probability that a standard deviation change in $C_g$ causes an increase in house prices higher than 2%.

**Sensitivity and robustness tests.**   From the DAG in Fig 2, we can infer most of our key assumptions, which are:

1. The effect of confounders on crimes shown on police.uk does not change over time $P(C_g|U, T) = P(C_g|U)$

2. The effect of confounders on potential nearby snaps does not change over time $P(M_s|U,T) = P(M_s|U)$

3. The distribution of confounders do not change over time $P(U|T) = P(U)$

4. The distribution of selling prices did not change for other reasons. To account for rising average house prices, we can demean $Y$ (or add an intercept term in regression models). For RQ2, we need the additional assumption:

5. Between $T = 0$ and $T = 1$, any change in the effect of $U$ on $Y$ is entirely mediated by the real crime count $C_r$.

Many of these assumptions involve $U$: common causes of $M_s$, $C_g$ and $Y$. Confounders can be split into two groups: observable and unobservable. Observable confounders are factors the research team has information about. In contrast, unobserved confounders are factors that the research team has no information on either because that information is inaccessible or the team is unaware of their existence.

For observable confounders, we can check for changes over time (assumptions 1–3) by comparing univariate and multivariate statistics between $T = 0$ and $T = 1$. For assumption 1, we can use an F-test comparing linear models:

$$C_g = \beta_{n0} + \beta_{nt}T + \beta_{nu}U + e_n \text{ (null model)}$$

$$C_g = \beta_{a0} + \beta_{at}T + \beta_{au}U + \beta_{ut}T * U + e_a \text{(alternative model)}$$

For demonstration, we specify $U$ as a continuous variable, but it can be categorical. Under the null model, the relationship between $U$ and $C_g$ remains the same over time (except for a scale shift accounted for by $T$). Under the alternative model, changes in the relationship between $U$ and $C_g$ are modelled as an interaction term. If an F-test rejects the null hypothesis that null model and the alternative model are equivalent, then assumption 1 is not credible. We test assumption 2 in the same way. To test assumption 3, we can use either a Fisher's exact test or a Kurskal-Wallis test, depending on whether $U$ is continuous or categorical.

In theory, any variable can be a confounder (unless proven otherwise). However, the most important confounders are the three inputs that determine what is shown on police.uk crime maps (see S2 File):

- the location that is queried on the crime map. In our case, this is the area around a property.

- the secret snap-point database

- police force recorded crime data

In our study, the research team can set the snap-point database used in the control periods (satisfying assumptions 1–3). For the location of houses and crimes, we can use geocoded information for tests (e.g. coordinates, higher areal units such as neighbourhoods). We can also use areal characteristics such as deprivation or access to amenities (see data section).

To test for changes over time for unobserved confounders, we can do a pre-intervention test. First, we find another period $T = -1$ (e.g. year 2009) before $T = 0$ where police.uk did not launch. Then we check that the same assumptions are met (e.g. $P(U|T = -1) = P(U|T = 0)$). Then we check if our estimators (e.g. 1A - 1C) give the expected result of no effects for years before any intervention took place.

We have no statistical way to test the assumption that between $T = 0$ and $T = 1$, any change in the effect of $U$ on $Y$ will be entirely mediated by the real crime count $C_r$. There may be other mediating pathways that do not travel along $U \rightarrow C_r \rightarrow Y$. For example. $Md$ is a mediator whose effect on house prices $Y$ differs between $T = 1$ and $T = 0$. If there is a single causal pathway between $Md$ and $Y$ that does not lie along the causal pathway $C_r \rightarrow M \rightarrow Y$ then this would violate our assumptions.

We do not expect spatial autocorrelation to affect our results greatly. However, we will test for spatial autocorrelation (i.e. checking Moran's I) and adjust estimates accordingly.

Finally, we can resort to investigative work to uncover evidence that may refute any of our assumptions. In particular, we will look for other ways that the police.uk crime maps may have affected house prices in non-crime related ways.

## Data

**Data sources / datasets used.**    The HM Land Registry Price Paid dataset is a publicly available dataset of properties sold in England and Wales since 1995. The dataset contains information on price sold, address, house type and other features of the property sold. The dataset excludes certain types of transactions, such as inheritance and discounted transactions (e.g. discounted sales of social housing under the 'Right to Buy' scheme). The dataset can be accessed at: https://www.gov.uk/government/collections/price-paid-data. The coordinates of a property are derived from the coordinates of its postcode as recorded in the ONS National Statistics Postcode Lookup [24].

Archival data from police.uk are publicly available from the police.uk data site [21]. Other information, such as police force boundaries, are also contained on the website. We use the earliest archival extract of police.uk which contains data on crimes from December 2010 to December 2013. In general, police.uk keeps excellent documentation on archival data and changes made to its website and data manipulation. We also cross-referenced the historical police.uk website using the Wayback machine, which is an archive of websites.

For SYP crime data, we will use the same data source sent to the Home Office and ultimately processed by police.uk. We expect data from at least the year 2010 to be available.

We infer the snaps used by police.uk from the unique crime locations shown on police.uk during these periods covered by versions one and two (see S1 and S3 Files). In our inferred dataset, we have 734,000 snaps in version 2, roughly 96% of all the snaps in use by police.uk during this period. The inferred snaps are much lower in version one (~462k); we do not know how many snaps were used in this version. Ideally, we would like to use the real snap-point database to mitigate against all measurement errors. However, for our estimators, it only matters that the causal relationship between our inferred snap-points and confounders remains constant during the period under study.

To test the plausibility of the research design, we use a variety of data sources to check for confounders. We aim to use a kitchen-sink approach: assume that any variable can be a confounder and test it all. Potential confounders include:

- Food agency standards ratings. This is point data that contain information on business type, address, latitude and longitude alongside the most recent hygiene rating of an establishment

- Radon readings per 1km square grid. This is dataset is publicly available.

- Aggregated data (LSOA) on households from the British Household panel

- Access to Healthy Assets and Hazards. This is data from the Consumer Data Research Centre (CDRC). These are areal (LSOA) measures of how 'healthy' a neighbourhood including the mean distance to:–retail environments (e.g. fast food outlets)–health services (e.g. GP offices, hospitals etc)–physical environment (e.g. green space)–air quality

- Levels of homeownership in the area (output area, lower super output area)

**Data access restrictions.**    Police force data is not directly available for research. Researchers need to contact and negotiate access with individual police forces. Other data sources

mentioned are publicly available or else require a free registered account. In the case of Ordinance Survey product (OS), researchers may need to purchase data from OS.

We inferred the master list of snap-points from public domain data from data.police.uk [22]. The inferred list cannot be used for reverse geomasking (i.e. to reveal the exact location of crimes and identify victims). We have explained exactly how to recreate this list using public data (see S3 File) and have shared our code on Github (https://github.com/MengLeZhang/crimeMaps-preReg-code).

**Variables/ codebook.**   The main variables used are:

police.uk variables (points data)

- Date of offence/ incident (truncated to year and month only)

- Home Office Offence Code

- Latitude and Longitude (WGS84, to be converted to Easting and Northing OSGB36)

   SYP Police force variables (to be confirmed)

- Date of offence/ incident

- Home Office Offence Code

- Easting and Northing (OSGB36)

- Other contexts (e.g. free text information about location)

   Land registry price paid data

- Date of transfer. Date when the sale was completed, as stated on the transfer deed.

- Price paid (in GBP)

- Postcode (joined to coordinates of postcode centroid via the ONS master postcode lookup)

- Type of property (e.g. Flats, Detached housing etc)

- Old/New. Whether a property was newly built or an established residential building

- Freehold or Leasehold Price Paid Data (PPD) category. Relates to type of price paid data and data recording changes over time. We use category A which forms the bulk of the dataset and is available from 1995. Category B transactions are only recorded since October 2013. These include transfers under a power of sale/repossessions and buy-to-lets.

   For every residential property sold, we can derive the following variables:

- number of potential snap-points nearby (based on a particular snap list version)

- number of crimes and incidents nearby in the past three months (police force recorded).

- number of crimes and incidents nearby in the past three months (police.uk recorded)

- number of crimes and incidents nearby in the past three months of available police.uk data.

There is a lag in police.uk data, homebuyers buying in February can only access data up to January (and maybe even less recently than that).

Data on crimes in the prior three months is chosen based on Braakman's research [14]. Nearby is defined as within 150, 300 or 500m; our preferred distance is 150 because the lowest level of points data on police.uk appears at a specific zoom level. At that zoom level, the scope of the interactive map on police.uk roughly covers a 300m by 300m square.

**Data quality issues.** From speaking to SYP, there are some data quality issues in the raw police geocoded crime and incidents data. First, some crimes and incidents will have no locations recorded, or locations are misrecorded. For instance, incidents with unknown or ambiguous locations are often recorded as taking place within police stations. Second, the data received by police.uk each month is a snapshot of police systems. The police continually update these records (e.g., omitting duplicate incidents), but these updates will not be reflected on public crime maps. For RQ2, our police records will be more up-to-date than those used to produce the police.uk crime maps in the past. The extent of these errors is unlikely to affect our results.

Aside from these issues, there are no missing values in our data. Where missing values exist, we will perform list-wise deletion (i.e. get rid of cases with missing fields). Public domain data on housing and crimes already have undergone data cleaning and error checks by their respective data owners. We will conduct checks on the SYP data. This is in addition to any data cleaning already done by the police.

To check that we can replicate police.uk's crime maps, we have cross-referenced statistics from our inferred snap-point list with the statistics from the real snap-point data (see S3 File). We will also use the raw police data to check that we can recreate the public crime data from 2011–2013.

## Ethics

This project has been approved by the University Research Ethics Committee at the University of Sheffield (approved 13/10/2021, reference no. 043654). The corresponding author submitted a University Research Ethics Committee-approved self-declaration to the ethics committee. This study involves no primary data collection or human participants. With the exception of South Yorkshire Police data, we exclusively use data released to the public domain by the UK government or other public bodies under the Open Government Licence v3.0 (or a similar licence) which permits use for research purposes. We have obtained written consent from South Yorkshire Police to use their data. The police data has been anonymised, and contains no personal information about offenders and victims. A full ethical review was waived since the research was judged to involve only existing data that has been robustly anonymised, and is unlikely to cause offence to those who originally provided the data. Our ethics approval letter is contained in the supplement.

## Project timeline

The project idea was conceived in 2020. An early pilot of the method for detecting snap-points was trialed using West Midlands Police data in December 2020. This pilot did not use any housing data. This project was funded in September 2021. The project aims and our hypotheses have not changed since initial funding. We used public domain data from South Yorkshire to test our methods in December 2021 (see S1 File). This revealed initial results for only one police force (out of 43 police forces) for RQ1. The study was pre-registered in April 2022. We are blind to the study's results and will commence data analysis in September 2022.

## Discussion

### Study limitations and risks

Police.uk's snaps database remains a secret and this guarantees that the snap point database cannot causally affect house prices except via police.uk's website and API. However, we have to replicate the snap-point database using public information. We can infer the majority of the

snap-points through police.uk data but our inferred database can still contain errors. We can test the accuracy of our inferred snap database using real police force data.

The scope of this study is only limited to a number of years, mainly 2010–2013. For RQ2, we can only study the effects for South Yorkshire.

Most data is within the public domain, except for SYP data. We have kept in close contact with SYP to minimise the risk of project termination due to the withdrawal of data access. In case SYP is unable to extract historical data, we will have to amend our statistical analysis or pursure RQ1 only.

## Data protection

A data protection plan formed between SYP and Sheffield University minimises the disclosure of personal data. All personal data will be stored within SYP setting. No personal data will leave these settings. All other data is publicly available. Only aggregated results will be taken out of the setting and cleared by SYP beforehand.

## Supporting information

**S1 File. Example of analysis using South Yorkshire.**
(DOCX)

**S2 File. Full DAG.**
(DOCX)

**S3 File. Materials related to police-uk.**
(DOCX)

## Acknowledgments

We thank Jamie Smith from South Yorkshire Police for helping us gain access to the data and addressing our numerous questions about the police data. We also thank Alberto Hidalgo for his comments and feedback on this protocol. We thank our colleague, Kitty Lymperopoulou, who collaborated with us on an earlier scoping project that eventually led to this study. We are indebted to her for her time. We contacted Lisa Tompson for statistics from her paper to do power size calculations before this project began. We are grateful for her help. This protocol uses data from OpenStreetMap (© OpenStreetMap contributors).

## Author Contributions

**Conceptualization:** Meng Le Zhang, Monsuru Adepeju.

**Data curation:** Meng Le Zhang, Rhiannon Thomas.

**Formal analysis:** Meng Le Zhang, Rhiannon Thomas.

**Funding acquisition:** Meng Le Zhang, Monsuru Adepeju.

**Investigation:** Meng Le Zhang, Monsuru Adepeju, Rhiannon Thomas.

**Methodology:** Meng Le Zhang.

**Project administration:** Meng Le Zhang.

**Supervision:** Meng Le Zhang.

**Visualization:** Meng Le Zhang, Rhiannon Thomas.

**Writing – original draft:** Meng Le Zhang.

**Writing – review & editing:** Meng Le Zhang, Monsuru Adepeju, Rhiannon Thomas.

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
