## [Decision Letter · Decision Letter 0]

26 Jul 2022

PONE-D-22-13559Estimating the Effects of Crime Maps on House Prices using an (Un)natural Experiment: A Study ProtocolPLOS ONE

Dear Dr. Zhang,

Thank you for submitting your manuscript to PLOS ONE. After careful consideration, we feel that it has merit but does not fully meet PLOS ONE’s publication criteria as it currently stands. Therefore, we invite you to submit a revised version of the manuscript that addresses the points raised during the review process.

We look forward to receiving your revised manuscript.

Kind regards,

Lubos Buzna, Ph.D

Academic Editor

PLOS ONE

Journal Requirements:

3. Please upload a new copy of Figure 1 as the detail is not clear. Please follow the link for more information: https://blogs.plos.org/plos/2019/06/looking-good-tips-for-creating-your-plos-figures-graphics/

4. We note you have included a table to which you do not refer in the text of your manuscript. Please ensure that you refer to Table 1 in your text; if accepted, production will need this reference to link the reader to the Table.

5. Please ensure that you refer to Figure 1 and 3 in your text as, if accepted, production will need this reference to link the reader to the figure.

6. Please upload a copy of Figure 7 and 5, to which you refer in your text on page 3 and 14. If the figure is no longer to be included as part of the submission please remove all reference to it within the text.

Reviewers' comments:

Reviewer's Responses to Questions

**Comments to the Author**

1. Does the manuscript provide a valid rationale for the proposed study, with clearly identified and justified research questions?

Reviewer #1: Yes

Reviewer #2: No

2. Is the protocol technically sound and planned in a manner that will lead to a meaningful outcome and allow testing the stated hypotheses?

Reviewer #1: Yes

Reviewer #2: No

3. Is the methodology feasible and described in sufficient detail to allow the work to be replicable?

Reviewer #1: Yes

Reviewer #2: No

4. Have the authors described where all data underlying the findings will be made available when the study is complete?

Reviewer #1: Yes

Reviewer #2: Yes

5. Is the manuscript presented in an intelligible fashion and written in standard English?

Reviewer #1: Yes

Reviewer #2: Yes

6. Review Comments to the Author

You may also provide optional suggestions and comments to authors that they might find helpful in planning their study.

Reviewer #1: I find the text a bit repetitive. Try to remove some ideas that are mentioned twice.

Page 16. In general, confounders are not properly defined. This is unclear.

Page 16. The definition of "I" and three types of "I" is unclear. How is this used later in the text?

Page 20. The text mentions Figure 5, which is not located in the main text.

Page 22. "share" should be "shared"

Table 1 column headings should be checked (Ln vs. Log)?

Table 1 should contain currency (the reader assumes this is GBP, right?).

The bibliography list is not homgeneous. The authors should mention how to cope with the risk of finding errors in the data, having solid data for such a project is not only related the presence of missing data.

The quality of preparation of this manuscript should increase.

Reviewer #2: The present paper gives an estimation of the causal effect of public crime maps on house prices. Based on the public crime map website www.police.uk , the authors conducted an experiment where the main objective is to specify the research plan ahead of data collection/Access and tried to answer two main questions: 1) Did police.uk crime statistics affect property prices? and 2) what is the effect of a one-unit increase in crime around a house (as reported by crime maps) on its selling price?

Whereas I find the results useful, the paper is very flawed in its structure, groundwork such as literature and data referencing and lacks a discussion section completely. It does not fulfil the format required by the journal, I think that the authors have disregarded the formal requirements and guidelines of the PLoS ONE journal. The manuscript organization is a bit confusing, for instance, sections are unnumbered, data sources appear at the end, the results section is located in the supplementary material and the findings and conclusions/discussions are missed. I suggest a section with a main heading and subsections up to 2-3 heading levels.

Below I give a (non-exhaustive) list of limitations of the paper, by section

1) Background

1.1. The research questions are clear however the authors mention that the early results are in the supplementary materials. Why?

1.2. Also, is hard to understand the meaning of the sentence: “If data access is granted before this protocol is finalised, we will stop any house price data (i.e. outcomes) from being analysed alongside the police force data.”

1.3. The authors state that RQ1 was answered by serendipitous discovery. What is the meaning of this?

2) Previous studies

2.1. Please stick to the correct reference style, even for data references and web pages.

2.2. The literature work is a bit sloppy and scarce. The authors state that: “no other study has studied the effect of crime maps on house prices or leveraged the type of design developed in this study.” However, other works that have studied the effect of crime maps on house prices and not referenced are, for instance:

- Dubin, R. A., & Goodman, A. C. (1982). Valuation of education and crime neighborhood characteristics through Hedonic Housing Prices. Population and Environment, 5, 166–181.

- Ceccato, V., & Wilhelmsson, M. (2012). Acts of vandalism and fear in neighbourhoods: Do they affect housing prices? In V. Ceccato (Ed.), The urban fabric of crime and fear (pp. 191–215). New York, London: Springer.

- Ceccato, V., & Wilhelmsson, M. (2018). Does crime impact real estate prices? An assessment of accessibility and location. J. Gerben, N. Bruinsma. & S. D. Johnson, (Eds.), Oxford Handbook on Environmental Criminology (pp. 518–544). Oxford University Press.

- Ceccato, V. & Wilhelmsson, M. (2020) Do crime hot spots affect housing prices? Nordic Journal of Criminology, 21:1, 84-102, DOI: 10.1080/2578983X.2019.1662595

- Buonanno, P., Montolio, D., & Raya-vílchez, J. M. (2012). Housing prices and crime perception. Empirical Economics, 45, 305–321

- Lynch, A. K., & Rasmussen, D. W. (2001). Measuring the impact of crime on house prices. Applied Economics, 33, 1981–1989

- Naroff, J. L., Hellman, D., & Skinner, D. (1980). Estimates of the impact of crime on property values. Growth and Change, 11, 24–30.

- Wilhelmsson, M., & Ceccato, V. (2015). Does burglary affect property prices in a nonmetropolitan municipality? Journal of Rural Studies, 39, 210–218.

3) Hypothesis

3.1. H1 is the hypothesis rejected by the serendipitous Discovery, whereas the authors state that H2 hypothesis has not been tested yet because they do not have extracted the data. If so, RQ2 remains unanswered and the research only has one research question. Please, clarify this.

3.2. Again, please stick to the correct reference style, even for data references and web pages

4) Research Design

4.1. What did the authors mean when they refer to “confounding (common causes)”?

4.2. Figure 2 is a bit confusing as estimated BetaT is an estimated parameter grouping several associations as correlation coefficient, regression slope, etc…

5) Methodology and Statistical Analysis

5.1. Please state more clearly, which period and periodicity are covered by your data. Is a bit confusing to understand which years are the data covering especially after reading the section Sample.

5.2. In the manuscript, there are continuous references to the supplementary material however none of these references are accurate, i.e. A1, A2, A3,…

5.3. Has been multicollinearity checked? The DAG resulting in the supplementary material reveals causal relations that may produce a correlation between independent variables.

6) Estimator RQ1

6.1. To check the results we had to move to the supplementary material.

6.2. Are the results statistically significant?

7) Estimator RQ2

7.1. Apparently, there are no results for Research Question 2. Neither in the manuscript nor in the supplementary material. If there is no outcomes, what is the point to include the RQ2? Please, clarify this.

8) Sensitivity and Robustness tests

8.1. Which tests for sensitivity and robustness did the authors applied?

9) Data sources

9.1. The authors mention a kitchen-sink approach for the impact of the confounders the role of the variable might be different: categorical, ordinal, quantitative, qualitative,… how did they deal with all of them?

It would have been interesting to determine what is the type of crime that most affects prices (vandalism, burglary, robbery, violent crime, etc…) even the kind of predominant population in the area, whether the…?

I completely miss a discussion of the main findings that goes deeper on what we learned from the paper regarding whether crime maps have an impact on house prices moreover since RQ2 has not been answered.

Overall, I think the results could be useful for further analysis. The paper, however, in its current form reads more like a (rather unstructured) analysis of the relationship between crime maps and house prices than an original study protocol. Conclusions are not presented in an appropriate fashion and the paper would need serious restructuring. In the current form, it is not acceptable for publication in this journal.

7. PLOS authors have the option to publish the peer review history of their article (what does this mean?). If published, this will include your full peer review and any attached files.

Reviewer #1: No

Reviewer #2: No

---

## [Author Response · Author response to Decision Letter 0]

9 Sep 2022

Response to reviewers (also see attached pdf for better formatted version) 

We would like to thank the reviewers and editors for their time in providing their comments; many of which

were detailed and specific. The majority of these from Reviewer 1 and the journal editorial team were about

the preparation of the manuscript. In response, we have made these major changes:

• Changed the document to conform with PLoSone guidelines in line with the editor’s suggestions

• Changed section heading to be more in line with PLoSone’s guidelines for protocols

• Uploaded figures up to standard of PLOSone using the PACE tool.

• Supplements are now correctly referenced.

• Added Ethics and funding letter to the supporting information (not for peer review). Improved ethic

subsection in the methods section in line with PLoSone guidelines.

• Clarification about confounders with examples in the statistical analysis subsection. We have also

clarified what we mean by observed and unobserved confounders (i.e. variables that the research team

can observe vs unobserable/ unavailable variables) in the sensitivity and robustness section.

• Added more references to studies on the effects of crime maps on public perception.

• Added a project timeline subsection to clarify the project status and the degree to which we are blind

to the results before starting data extraction and analysis.

• Cut down on text repetition by moving the technical details of snap points and geomasking to supplement

S3. The main text no longer repeats the core design ideas as much.

• Improved discussion of data errors. We have added in additional information from SYP on the raw

police data. Subsection has been renamed data quality instead of missing data to reflect change.

• Fixed numerous minor issues (typos, figure referencing, table heading, referencing tables in the text etc)

We note that reviewer 1 is positive in their review (answering yes to all check items). Reviewer 2 has a

completely different opinion. We believe the majority of reviewer 2’s comments stem from a misunderstanding:

this is a study protocol and not a full report on findings. The definition of a study protocol from PLOSOne:

Study Protocols describe detailed plans for conducting research, including the background,

rationale, objectives, methodology, statistical plan, and organization of a research project. PLOS

ONE accepts submissions of Study Protocols for any study type within the journal’s scope.

However, Reviewer two has valid criticisms which we have addressed (see itemised responses). However, we

want to address reviewer two criticism of our paper’s originality here.

Reviewer two believes the literature work is sparse; truthfully that’s because the literature on the impact of

crime maps is scarce. However, we have included additional references on related studies looking at public

perceptions of crime maps. They also challenged our assertion: “no other study has studied the effect of

crime maps on house prices or leveraged the type of design developed in this study.” Then provides a list of

reference.

We have fact-checked this references to check 1) the estimand of these studies (i.e. do they measure crime map

effects or something else) and 2) the design of the study (i.e. did they use geomasking errors or other quirks

of crime data to separate the effects of Open Data from other crime effects). First, none of them look at the

introduction of crime maps on house prices. They all look at relationship between crime in general on house

prices. This is not the same topic. Second, they almost all depend on statistical adjustments with no design

beyond throwing whatever covariates into a regression model (aside from those using instrumental variables

with implausible instruments). We rely on geomasking to provide nearly as good as random variations in crime

information available to the public (with interrupted time series taking out any remaining confounding bias).

Third, we already cite a systematic review of crime effects on house prices in our submission (Ihanfeldt and

Mayock) which also comes to the same conclusion about this literatures (e.g. weak identification strategies).

1

Our original comments about novelty still stand. The reviewers can check our summaries of each article

(under reply to reviewer 2 comments) to make up their own minds.

Regarding our current status. When we submitted the paper in May, we had not yet extracted any data in

order to remain blind to the final results. The only exception is public domain data from South Yorkshire

which was used to test code, scope statistical power, and demonstrate our estimators. Since we know the

early results from one police force (for RQ1 only), we have openly declared this. We have not used public

domain data from any of the other 40+ forces for the protocol (but will do so for the study). This is a project

with limited resources and time. We have since gained access to raw South Yorkshire Police data (used for

RQ2) and are poised to start analysing public domain data (used for RQ1) in this September (see project

timeline subsection).

We are happy for PLOSone to publish the history of this submission so that all readers can judge that we

have not changed our hypotheses and research questions between initial submission and resubmission (in fact

our hypotheses remain the same as those in our study pre-registration on OSF).

List of detailed responses

Editor comments

⊠ Please ensure that your manuscript meets PLOS ONE’s style requirements, including those for file

naming. The PLOS ONE style templates can be found at https://journals.plos.org/plosone/s/file?id

=wjVg/PLOSOne_formatting_sample_main_body.pdf and https://journals.plos.org/plosone/s/file?

id=ba62/PLOSOne_formatting_sample_title_authors_affiliations.pdf

• A: Changed as per requested. Title page formatted as per instruction. Figures and heading formatted

as per instruction.

⊠ Please include your full ethics statement in the ‘Methods’ section of your manuscript file. In your

statement, please include the full name of the IRB or ethics committee who approved or waived your

study, as well as whether or not you obtained informed written or verbal consent. If consent was waived

for your study, please include this information in your statement as well.

• A: Added and ethics letter is in supplement (S4).

⊠ Please upload a new copy of Figure 1 as the detail is not clear. Please follow the link for more information:

https://blogs.plos.org/plos/2019/06/looking-good-tips-for-creating-your-plos-figures-graphics/

• A: Done. All figures are converted and checked through PACE

⊠ We note you have included a table to which you do not refer in the text of your manuscript. Please

ensure that you refer to Table 1 in your text; if accepted, production will need this reference to link the

reader to the Table.

• A: Done

⊠ Please ensure that you refer to Figure 1 and 3 in your text as, if accepted, production will need this

reference to link the reader to the figure.

• A: references added in text

⊠ Please upload a copy of Figure 7 and 5, to which you refer in your text on page 3 and 14. If the figure

is no longer to be included as part of the submission please remove all reference to it within the text.

• A: Original figure 5 reference changed. Figure 7 is a figure in a referenced paper. We took out this

reference to a figure and left in a reference to the paper.

⊠ Please include captions for your Supporting Information files at the end of your manuscript,

and update any in-text citations to match accordingly. Please see our Supporting Information

2

guidelines for more information: http://journals.plos.org/plosone/s/supporting-information. A:

Done. File names changed and put captions at end of main text.

Review one comments

⊠ I find the text a bit repetitive. Try to remove some ideas that are mentioned twice.

• A: We have cut down the text and avoided repetition of the research design. We have moved the

technical details for the snap point and geomasking to Supplement S3 (now renamed to material related

to police-uk). This should greatly cut down repetition of key ideas.

⊠ Page 16. In general, confounders are not properly defined. This is unclear.

• A: We define the technical term confounders in the research design; this is common term used throughout

the social sciences and health related research. We have added a section with examples of confounder

in the statistical analysis section. We have gone into details in the sensitivity and robustness tests

and shown why the location of houses and crimes are the most important confounders in S2. We have

also clarified what we mean by observed and unobserved confounders in the sensitivity and robustness

section. We do reiterate that anything can be a confounder (unless proven otherwise).

⊠ Page 16. The definition of “I” and three types of “I” is unclear. How is this used later in the text?

• A : Deleted in main text. This definition is used in the S2.

⊠ Page 20. The text mentions Figure 5, which is not located in the main text.

• A: Figure 5 is amended to figure 3.

⊠ Page 22. “share” should be “shared”

• A: done

⊠ Table 1 column headings should be checked (Ln vs. Log)?

• A: Changed to log which is normally interpreted as the natural log (ln)

⊠ Table 1 should contain currency (the reader assumes this is GBP, right?).

• A: Done. References to GBP are also made in the text.

⊠ The bibliography list is not homgeneous.

• A: The bibliography list has been amended to Vancouver style as mandated by PLOSone.

⊠ The authors should mention how to cope with the risk of finding errors in the data, having solid data

for such a project is not only related to the presence of missing data.

• A: We have filled this out more extensively. The public domain data has already been error checked.

We have noted potential errors in the raw police data (which will be reflected in the public data). Based

on talking with the police and their crime mapper – these issues are unlikely to affect our results. We

have reiterated that we have checked that our Snap data is an accurate recreation of the real secret

snap data. We recreate the public domain data from 2011-2013 using raw police data to check that we

have recreated the police.uk data routine. Section has been renamed data quality instead of missing

data to reflect.

⊠ The quality of preparation of this manuscript should increase.

• A: Following the advice of the reviewer and the editor, we have amended the manuscript with reference

to PLOSone’s guidelines.

3

Reviewer two comments:

⊠ The manuscript organization is a bit confusing, for instance, sections are unnumbered, data sources

appear at the end, the results section is located in the supplementary material and the findings and

conclusions/discussions are missed . . . the paper is very flawed in its structure, groundwork such as

literature and data referencing and lacks a discussion section completely.

• A: This paper follows the structure of a PLOSone/ OSF protocol template. The original paper also did

so with some exceptions. Deviations from the standard format occured because we are doing secondary

data study using a potential natural experiment rather than conducting an actual experiment with new

data collection. The paper now follows the standard PLOSone protocol template even more rigidly.

We have amended the referencing. The discussion section was called Other Consideration in original

manuscript (now renamed). The discussion in a study protocol is not meant to be a discussion of results

(since there are no results yet). PLoSone does not mandate numbered sections. See example here:

Plante C, Bandara T, Baugh Littlejohns L, Sandhu N, Pham A, Neudorf C. Surveying the

local public health response to COVID-19 in Canada: Study protocol. Plos one. 2021 Nov

18;16(11):e0259590.

⊠ Background. Why is results in supplementary materials. what is the meaning of the data access

sentence. what does it mean by serendipitous discovery.

• A: As mentioned earlier, this is a study protocol and not a result paper. However, in S1 we show an

example analysis using 1 police force out of 43 to make it clear to the readers a) what we have already

done (to test our methods) and b) what the analysis looks like in practise (the statistical analysis

section is very abstract and dry). We have moved mention of the serendipitous discovery to the project

timelines subsection (in line with PLOSone protocol templates). This ought to clear things up.

⊠ H1 is the hypothesis rejected by the serendipitous Discovery, whereas the authors state that H2

hypothesis has not been tested yet because they do not have extracted the data. If so, RQ2 remains

unanswered and the research only has one research question. Please, clarify this.

• A: Please, see the above comment on the category of the current paper. However, we address the

confusion relating to the hypothesis in the study timeline. To test our methods for RQ1, we had to trial

our method on one police force. Our total sample consists of data from more than 40 forces. Hypothesis

1 has not been rejected but we are not completely blind to the results from one force.

⊠ hypothesis: correct reference style for data and webpage.

• A: Correct referencing

⊠ design. what are confounders (common causes).

• A: this is a common term in the social sciences (including economics) and health. We have explained it

in the research design and added examples in the statistical analysis section.

⊠ design. BetaT is used too much. Choose another letter

• A: We’ve kept the statistical notation. There’s a lot of formulas and we’ve sectioned off our estimators

in order to avoid confusion around what BetaT is from one section to another. BetaT is normally a

test statistic.

⊠ method. time period.

• A: We’ve moved the sample subsection to nearer the beginning of the material and methods. We’ve

made the time period as clear as possible. The end period is always the year 2013. The earliest year for

RQ2 is unknown (2010 being the latest it can be).

⊠ appendix refenreces not right???

• A: We’ve renamed supplements according to PLOSone guidelines.

⊠ collinearity

4

• A. Collinearity does not bias estimates unless it is particular bad (i.e. near perfect collinearity). This is

not the case; we’ve quoted Tompson et al who demonstrated there is a considerable mismatch between

police.uk reported crime and police force recorded crimes. Multiple covariate correlated with the

‘exposure’ (e.g. geomasking error) will reduces statistical power however as reviewer 2 can see all our

estimators either have no covariates or few covariates (2 in the case of RQ1). We have used South

Yorkshire data in supplement S1 to demonstrate power and noted other unknowns around power in the

main text.

⊠ RQ1. clarify to reviewer 2 re: study protocol,

• A. Please, see our earlier comment on the category of the current paper.

⊠ RQ2. See RQ1

• A. Please, see our earlier comment on the category of the current paper.

⊠ sensitivity and robustness. see RQ1 and 2

• A. Please, see our earlier comment on the category of the current paper. We have expanded the section

on robustness tests to give exact statistical tests. The worked example in S1 gives an idea of what we

want to do.

⊠ data source. what is the different tests for each one?

• A. We’ve stated the exact statistical tests in the robustness section in detail. These are all standard

univariate and multivariate tests.

⊠ Scope, more crime etc.

• A: A study protocol outlines the main research questions. It does not stop us from doing exploratory

results so long as we are clear what is confirmatory hypothesis testing (e.g. what’s in the protocol

written before results are known) and what are exploratory results. The scope of our main objectives is

to cover an area that is academically novel, within the public interest, and based on hypotheses formed

from arguments from the Government and private bodies such as estate agents.

Fact-check of papers

For comparison’s sake, we start with the estimand and research design of the current study:

This paper

Estimand: Effect of introducing public crime maps on house prices (RQ1). We also estimate how house prices

are affected when more crimes are shown in an area on a public crime map (RQ2).

Research design: Before and after analysis using interrupted time series. The design uses features of crime

map to isolate the effects. These are features (master snap list and extent of geomasking error) are secret and

kept from the public. Any remaining confounding is eliminated through the longitudinal design. Robustness

tests exist to test the assumption if confounder bias has been eliminated or render negligible. A protocol and

pre-registration (i.e. this current submission) is created to reduce opportunities to change the hypotheses,

design or statistical analysis after seeing the results. This is unusual for economics.

Dubin, R. A., & Goodman, A. C. (1982). Valuation of education and crime neighborhood

characteristics through Hedonic Housing Prices. Population and Environment, 5, 166–181.

Estimand: The causal estimand is not clearly stated per se but implicity it is the effect of crime in

neighbourhood on house prices. E.g. if a crime occured in neighbourhood what is its average effect on house

prices.

Research design: Regression modelling only. No design based exogenous source of variation. Bizzarely uses R

squared and sign of coefficients to assess the validity of results.

5

Note: This is a 1980s paper where the cutting edge for causal inferences in econometrics was weaker so our

review is overly harsh. The estimand is not about online crime maps or open data; this paper was published

before the internet age!

Ceccato, V., & Wilhelmsson, M. (2012). Acts of vandalism and fear in neighbourhoods: Do they

affect housing prices? In V. Ceccato (Ed.), The urban fabric of crime and fear (pp. 191–215).

New York, London: Springer.

Estimand: Crime (vandalism) on house prices.

Design: Statistical adjustment with no design based source of exogeneity expect for one instrumental variable

estimator. Murder is used as an instrument for vandalism; very little rationale is given aside from another

paper by Tita et al (2006). ‘According to Tita et al. (2006), murder is an ideal instrument’ is their only

narrative justification before moving onto statistical tests.

We investigate Tita et al (2006) to check for the rationale behind murder as an instrument to correct

measurement error (e.g. underreporting) in violent crimes. The rationale of the paper is not use murder to

solve other confounding issues with estimating the effects of

Ceccato, V., & Wilhelmsson, M. (2018). Does crime impact real estate prices? An assessment of

accessibility and location. J. Gerben, N. Bruinsma. & S. D. Johnson, (Eds.), Oxford Handbook

on Environmental Criminology (pp. 518–544). Oxford University Press.

Estimand: Crime on house prices.

Design: We could not access the chapter but it seems like statistically models without a design-based source

of exogeneity is used.

Ceccato, V. & Wilhelmsson, M. (2020) Do crime hot spots affect housing prices? Nordic

Journal of Criminology, 21:1, 84-102, DOI: 10.1080/2578983X.2019.1662595

Estimand: Crime (vandalism hotspots) on house prices

Design: Hedonic modelling with no design based source of exogeneity. Unlike their previous articles, the

authors chose not to implement IV estimation (lack of strong and credible instruents).

Buonanno, P., Montolio, D., & Raya-vílchez, J. M. (2012). Housing prices and crime perception.

Empirical Economics, 45, 305–321

Estimand: Crime perception on house prices. We find this one strange because actual crime is a cause of

both crime perceptions and other mediating factors (e.g. property damage) that affect house prices.

Design: Use of instrumental variables with historical levels of victismation and share of youths as instrument

for crime perception in the modern day. Other sources of confounding are adjusted for using regression

adjustment. Although we are not convinced about the credibility of the instruments, they do dedicate some

space to discuss their choice of instruments. Exogeneity of instruments is tested through the Sargan-Hansen’s

J- test which has flaws (see our summary of Wilhelmsson, M., & Ceccato 2015).

Lynch, A. K., & Rasmussen, D. W. (2001). Measuring the impact of crime on house prices.

Applied Economics, 33, 1981–1989

Estimand: Crime on house prices.

Design: Hedonic modelling with no design based source of exogeneity.

Naroff, J. L., Hellman, D., & Skinner, D. (1980). Estimates of the impact of crime on property

values. Growth and Change, 11, 24–30.

Estimand: Effect of crime on house prices.

Design: Unsure. We cannot access the full paper. But the estimator is likely to be a reduced form regression

equation like in Dubin and Goodman (1982). In which case the same comment apply re: lack of design based

exogeneity.

6

Wilhelmsson, M., & Ceccato, V. (2015). Does burglary affect property prices in a nonmetropolitan municipality? Journal of Rural Studies, 39, 210–218.

Estimand: Crime (in this case burgarly) on house prices.

Design: Statistical adjustment only: hedonic modelling with alternative specifications for spatial autocorrelation and modelling quantiles. One estimator (out of several) has a design based approach: share of young

males in the area and the share of convenience stores are used instruments is used as an instrument for

burglary. Barely any rationale is given for the choice of instrument and a Sargan-Hansen’s J- test is used to

test the instrument validity. This is a weak test that will pass dubious instruments:

such tests reject instrument validity may often barely exceed small levels, even when instruments

are seriously invalid, whereas even minor invalidity of instruments can severely undermine inference

on regression coefficients by instrumental variable estimators. These uncomfortable patterns may

be aggravated when particular valid or invalid instruments are relatively weak or strong.

From:

Kiviet, Jan F., and Sebastian Kripfganz. 2021. ‘Instrument Approval by the Sargan Test

and Its Consequences for Coefficient Estimation’. Economics Letters 205 (August): 109935.

https://doi.org/10.1016/j.econlet.2021.109935.

---

## [Decision Letter · Decision Letter 1]

17 Nov 2022

Estimating the Effects of Crime Maps on House Prices using an (Un)natural Experiment: A Study Protocol

PONE-D-22-13559R1

Dear Dr. Zhang,

We’re pleased to inform you that your manuscript has been judged scientifically suitable for publication and will be formally accepted for publication once it meets all outstanding technical requirements.

Kind regards,

Lubos Buzna, Ph.D

Academic Editor

PLOS ONE

Additional Editor Comments (optional):

Thank you for amending the text of the manuscript. Both reviewers are fully satisfied and consider the paper to be publishable. Please pay some attention to the final remark made by reviewer 2. Congratulations to the paper acceptance.

Reviewers' comments:

Reviewer's Responses to Questions

**Comments to the Author**

1. Does the manuscript provide a valid rationale for the proposed study, with clearly identified and justified research questions?

Reviewer #1: Yes

Reviewer #2: Yes

2. Is the protocol technically sound and planned in a manner that will lead to a meaningful outcome and allow testing the stated hypotheses?

Reviewer #1: Yes

Reviewer #2: Yes

3. Is the methodology feasible and described in sufficient detail to allow the work to be replicable?

Reviewer #1: Yes

Reviewer #2: Yes

4. Have the authors described where all data underlying the findings will be made available when the study is complete?

Reviewer #1: Yes

Reviewer #2: Yes

5. Is the manuscript presented in an intelligible fashion and written in standard English?

Reviewer #1: Yes

Reviewer #2: Yes

6. Review Comments to the Author

You may also provide optional suggestions and comments to authors that they might find helpful in planning their study.

Reviewer #1: No more comments. I already provided some suggestions in the first round and I am fully satisfied with the replies.

Reviewer #2: First of all, the authors made a substantial improvement in the manuscript. Now it reads better and formally is more suitable for its publication. However several remarks on the comments from the authors due to the review.

As reviewer I do not care about what other reviews do, whether they are positive or not. My only concern is to review in a proper way which implies profesionalism, criticism, objectivity and imparciality. Contrasting and ensuring that all the requirements and requisites either article, protocol or whatever the research is has been done rigurously and if not whether it can be improved.

I am perfectly aware of what a study protocol is, therefore is not a misunderstanding. Thus, the comments were done according to several aspects which I missed in this type of research. Apart of the formatting and journal requirements, which its lack was, undoubtedly, evident, the criticism on literature review is funded on the basis that although the references on “crime mapping effects (or impact) on house prices” may be scarce, what is not scarce is the literature on “crime mapping” and this is something that authors may be aware of. Also the criticism on the assertion “no other study has studied the effect of crime maps on house prices or leveraged the type of design developed in this study” should not be taken as a challenge nor the references provided. This is done to show the authors that the literature review may not be as deep as expected. In fact, the following reference is on the same topic and by the way it shows that spatial autocorrelation effects exist in terms of house price although there is a lack of understanding on how such spatial effects impact on the property market (the authors state that they do not expect that spatial autocorrelation to affect their results greatly:

David McIlhatton William Stanley McGreal Paloma Taltavul de la Paz Alastair Adair , (2016),"Impact of crime on spatial analysis of house prices: evidence from a UK city", International Journal of Housing Markets and Analysis, Vol. 9 Iss 4 pp. – http://dx.doi.org/10.1108/IJHMA-10-2015-0065.

Apart from that, I do insist that the authors have done a good work correcting and modifying the suggestions thus, the manuscript has improved sufficiently to be considered as publishable.

7. PLOS authors have the option to publish the peer review history of their article (what does this mean?). If published, this will include your full peer review and any attached files.

Reviewer #1: No

Reviewer #2: No

---

## [Editor Report · Acceptance letter]

18 Nov 2022

PONE-D-22-13559R1 

Estimating the Effects of Crime Maps on House Prices using an (Un)natural Experiment: A Study Protocol 

Dear Dr. Zhang:

I'm pleased to inform you that your manuscript has been deemed suitable for publication in PLOS ONE. Congratulations! Your manuscript is now with our production department. 

Kind regards, 

on behalf of

Prof. Lubos Buzna 

Academic Editor

PLOS ONE